# Surface Finishing of 3D-Printed Polymers with Selected Coatings

**DOI:** 10.3390/polym12122797

**Published:** 2020-11-26

**Authors:** Jure Žigon, Mirko Kariž, Matjaž Pavlič

**Affiliations:** Department of Wood Science and Technology, Biotechnical Faculty, Jamnikarjeva ulica 101, University of Ljubljana, 1000 Ljubljana, Slovenia; mirko.kariz@bf.uni-lj.si (M.K.); matjaz.pavlic@bf.uni-lj.si (M.P.)

**Keywords:** acrylonitrile-butadiene-styrene, coating, fused deposition modeling, poly(lactic acid), polymers, surface, wood

## Abstract

Surface treatment of 3D-printed objects with coatings, besides protection against environmental influences, offers the improvement of visual appearance of the printed elements. In order to design an optimum surface system, the physical and chemical properties of polymers surfaces should be well-known. In the present study, 3D-printed samples of acrylonitrile-butadiene-styrene, poly(lactic acid) and poly(lactic acid) with wood flour additive were coated with three different types of coating, namely solvent borne alkyd coating, water borne acrylic coating and coating made of acrylonitrile-butadiene-styrene diluted in acetone. The surface properties of substrates and the properties of surface systems were assessed with different methods. The results revealed the surfaces of polymers having hydrophobic character, whereas the color, gloss, surface roughness and coating film thickness of coated surfaces depend on the characteristics of particular coatings. Finally, the adhesion of coatings was shown to be appropriate, but dependent on substrate surface porosity and chemical properties of both substrate surface and coating asset.

## 1. Introduction

Three-dimensional (3D) printing presents an additive manufacturing process [1]. It enables the production of complex-shaped objects on the basis of 3D computer models [2,3]. Nowadays, the most common technology for producing a 3D-printed objects is a fused deposition modeling (FDM) [4]. FDM printer heats the tip of a thermoplastic polymer filament to a molten state and deposits it in successive layers to form the desired object [5,6]. The raw materials for filaments, which are supplied to the extruder and nozzle of 3D printer during the printing process, can be made of petroleum-based polymers or bio-based polymers [7]. In order to improve the desired properties of the filaments or reduce their price, different additives can be added [8], like for instance wood flour [9,10,11]. Surface roughness and wettability of the 3D-printed products are significant factors affecting their ability to be finished with different kinds of coatings [12].

3D-printed products can be used in different applications and are consequently exposed to different environmental conditions. When used in harsher conditions, for instance with elevated relative humidity and ultra-violet light irradiation, this drives the need for surface treatment with coatings that would offer both protection and desired aesthetic appearance of 3D-printed products [13]. The creation of a suitable surface system composed of polymers surface and coating asset requires a thorough understanding of the surface properties, surface chemistry and surface wettability, in particular, since these properties directly relate to coatings adhesion to the substrates. On the other hand, in order to design an optimum surface system, the physical and chemical properties of coatings should also be well-known.

Due to specific surface properties, the surface treatment of polymers is possible only with proper coatings. Indeed, such coatings usually contain different kinds of resins (acrylics, alkyd, polyurethane, polyvinylacetate, melamine, polyester) and solvents (water, butyl-alcohol, isopropyl alcohol, propylene glycol methyl ether) [14,15]. In addition, the coatings can be applied to 3D-printed objects with various technologies, depending on the coating type and properties, object shape and object size [16].

However, hydrophobic nature and other specific surface properties of polymers for 3D-printing make the surface treatment of printed objects with coatings quite challenging [17,18]. The physical, chemical and morphological properties of printing polymers, even when these contain wood fillers [19] and applied coatings, can significantly influence the interaction (including wettability and adhesion) between both materials [20]. A key issue of layer by layer manufacturing is also a high surface roughness of 3D-printed products. Surface finishing of manufactured objects can be controlled and improved by various pre- and post-processing methods, including plasma treatment, laser micro machining, chemical treatments and sanding [21,22].

To date, reports on surface treatment of 3D-printed objects, printed with various raw materials and coated with various assets and techniques can be found in the literature. In the following text, the findings from selected publications, mainly related to the present research, are described.

Acrylonitrile-butadiene-styrene (ABS) is an important engineering thermoplastic copolymer of acrylonitrile (A), butadiene (B), and styrene (S). ABS is widely used in industry due to its superior physical and mechanical properties, chemical resistance, ease of processing, affordability and recyclability [23,24]. ABS is also commonly used to make wood plastic composites [25]. Superhydrophobic coatings made of ABS were produced and investigated by Deng and co-authors [26]. Bateni and co-workers [27] coated natural fibers with ABS and successfully protected fibers from water absorption and decreased the biodegradation potential of the fibers in contact with soil.

Poly(lactic acid) (PLA) is a polymer with many advantageous properties. It is produced from natural feedstock, is compostable, and has a good stiffness and strength [28]. Additionally, due to biocompatibility, it can be well combined with wood and other natural fibers [18], which offers production of a new PLA-wood (PLA-W) composites with enhanced physicomechanical properties [29,30]. Kowalczyk and co-workers [31] increased the hydrophilicity of PLA by polyvinylpyrrolidone grafting onto polymeric substrate. On the other hand, Lee and co-authors [32] managed to prepare superhydrophobic PLA patterned surface structures and via dip coating process with application of silica nanoparticles and methyl ethyl ketone.

There is still a lack of publications dealing with the properties of 3D-printed products, surface treated with typical resin-based coatings. The aim of the present research was to assess the properties of three different substrates, 3D-printed of typical raw materials used for such purposes (ABS, PLA and PLA-W). The substrates were subsequently coated with three different coatings, each containing a different type of solvent. The hypothesis was that surface physical, morphological and visual properties of 3D-printed products depend on substrate type and a type of applied coating. It was additionally assumed that adhesion strength between the substrate and coating film depends on the type of both surface system components. Firstly, the wettability properties of the substrates were determined, and chemical properties of substrates and coatings were investigated. After coating application, the visual properties of coated samples were assessed, including color, gloss and surface roughness. In order to investigate the interface areas between the substrates and coating, to determine the coating film thicknesses, and to support certain findings of the study, a microscopic analysis of the samples was performed. Finally, the adhesion strength of the coatings films on the substrates was determined with pull-off test.

## 2. Materials and Methods

The methodology and analyses included in the study are presented in Figure 1 and described in the following subsections.

### 2.1. 3D Printing of the Substrates

Commercial filaments with 1.75 mm diameter were used for 3D printing. ABS was supplied by Zortrax (Olszytn, Poland) whereas PLA and PLA-W were supplied by local supplier (Plastika Trček d.o.o., Ljubljana, Slovenia). PLA-W filament is based on PLA polymer with addition of up to 40% (wt%) of pine wood particles.

Flat samples with dimensions of 150 × 75 × 4 mm were modeled with SolidWorks software (SolidWorks Corp., Massachusetts, USA), and STL model was sliced and prepared for 3D printing in Z-Suite (Zortrax, Olsztyn, Poland) software and manufactured on 3D-printer Zortrax M200 (Zortrax, Olsztyn, Poland). The layers with a thickness of 0.39 mm were formed by using the 0.6 mm nozzle. Due to the different melting temperatures of materials for 3D-printing, the processing temperatures were varied, in order to get the 3D-printed properties with optimal characteristics; the printing and bed temperature when printing ABS were set to 275 and 80 °C, whereas the printing and bed temperature when printing PLA and PLA-W were 210 and 30 °C. Two series of the substrates were used in the study: one series of the samples remained non-sanded and the other series was sanded with sanding paper grit P120.

### 2.2. Surface Wettability and Determination of Surface Free Energy (SFE)

The surface wetting analysis of non-sanded and sanded 3D-printed materials was performed by contact angle (CA) measurements of three test liquids: deionized water, diiodomethane and formamide. Optical goniometer Theta (Biolin Scientific Oy, Espoo, Finland) and the corresponding software (OneAttension version 2.4 (r4931), Biolin Scientific, Oy, Espoo, Finland) were used to measure the CA between the substrate’s surface and the tangent fitting to the shape of 3-µL droplet on both sides, by the Young–Laplace analysis [33,34]. Ten droplets of each liquid were applied on three 60 × 20 mm samples of individual substrate. CAs data of test liquids, obtained 2 s after droplets deposition, were used to calculate SFE according to the Owens–Wendt method [35].

### 2.3. Attenuated Total Reflection Fourier Transform Infrared (ATR FT-IR) Spectroscopy

ATR FT-IR spectroscopic measurements of the 3D-printed substrates and coatings were performed by using a Perkin Elmer Spectrum Two spectrometer (PerkinElmer Inc., Waltham, MA, USA). Three spectra per type of material were collected (16 scans per measurement) at a wavelength from 600 cm^−1^ to 4000 cm^−1^, at a resolution of 0.5 cm^−1^ in the transmittance mode.

### 2.4. Coatings and Coating Films Formation

Three different types of the coatings (two of them commercially available) were selected: waterborne acrylic coating Belinka exterier email (C-WB, Helios TBLUS d.o.o., Količevo, Slovenia), solvent borne alkyd coating Tessarol professional email (C-SB, Helios TBLUS d.o.o), and the coating made of ABS diluted in acetone (C-ABS, weight ratio 50% ABS: 50% acetone) [36,37].

Coatings were manually applied on 3D-printed samples with a quadruple coating applicator Erichsen model 360 (Erichsen GmbH & Co. KG., Hemer, Germany). The coatings were applied at the rate of applicator movement of 30 mm·s^−1^, forming the coating films with the wet film thickness of 240 μm. The coated samples were stored for 3 weeks in a room with temperature of 20 °C and relative humidity of 50% prior to further analyses.

### 2.5. Determination of Visual Properties

The visual properties, including gloss and color of the uncoated and coated samples’ surfaces were measured for each property on 5 spots on every sample. Gloss, as a property related to the intensity of light reflected in specular direction, was measured with a gloss meter (X-Rite, AcuGloss TRI, Grand Rapids, MI, USA) by the angle between the light source and the observed plane of 60°. CIELAB color space components (*L**, *a** and *b**) were measured with the spectrophotometer X-Rite (Grand Rapids, MI, USA) SP62 with the D65 type of light.

### 2.6. Determination of Surface Roughness

Surface roughness of non-coated and coated 3D-printed samples was studied with confocal laser scanning microscope (CLSM) LEXT OLS5000 (Olympus, Tokyo, Japan). The analysis was performed on 5 different spots on samples’ surfaces. The area of the individual sample was observed with a laser light source with a wavelength of 405 nm at a maximum lateral resolution of 0.12 µm. The topographical images of the area were taken at 5-fold magnification (scanned area of about 2560 µm^2^), and the software OLS50-S-AA (Olympus, Tokyo, Japan) was used to calculate the areal roughness parameter *S*_a_ (arithmetic mean roughness).

### 2.7. Microscopic Analysis

The thickness of the dry coating film was measured at 20 spots on the cross-sections of coated samples, at 5-fold magnification, by using the light microscope module of the CLSM microscope.

Cross-section microscopic structure of sanded ABS coated C-ABS and microscopic structure of non-sanded PLA-W surface were analyzed with scanning electron microscope (SEM) FEI Quanta 250 (FEI, Hillsboro, OR, USA). Samples of dimensions 5 × 5 × 4 mm were sprayed with a gold conductive layer prior to observations. The analyses of the samples were performed in a high vacuum of 1.06e^-4^ Pa, the electron source voltage was 10.0 kV, and the spot size was 3.0 nm. The images were captured by the time of the beam transition through the sample of 45 μs.

### 2.8. Determination of Coating Adhesion Strength

Adhesion of the coatings on the substrates was evaluated by the pull-off tests, according to the standard EN ISO 4246 [38]. Aluminum dollies with a diameter of 20 mm were glued with a two-component epoxy resin Endfest plus 300 (UHU GmbH & Co. KG, Bühl, Germany) on the surface of coated specimens. Due to adhesive failure in case of samples coated with C-ABS, the dollies were furtherly glued with single-component cyanoacrylate glue Cianokol PRO (Mitol, Sežana, Slovenia). After 24 h of curing, the boundaries of the glued dollies were carefully cleaned down to the substrate, to prevent propagation of failures out of the tested area. The adhesion strengths of the cured films were measured by using the pull-off testing machine DeFelsko Positest adhesion tester (DeFelsko Corporation, Ogdensburg, NY, USA) until the separation of the dolly from the specimens’ surfaces occurred. If a separation between the substrate and the film occurs (at least 60%), the adhesion strength is determined; otherwise, the strength is considered as a cohesive one if the substrate failure is predominant.

## 3. Results and Discussion

### 3.1. Surface Free Energy

Previous studies reporting on wettability of ABS and PLA 3D-printed products concluded that the layer thickness of printing has the largest impact on the hydrophobic performance due to different surface roughness, followed by the filling method and the printing speed [39]. The same authors concluded that by adjusting the printing parameters, a good surface quality as a prerequisite before coating application can be achieved.

The calculated dispersive and polar part of SFE measured of different substrates, together with initial water CAs, are presented in Figure 2. The sanding of the surfaces did not influence calculated SFE and in some cases even slightly decreased it. SFE of all types of substrates mainly presented dispersive component and in average accounted to about 30 mJ∙m^−2^. The majority of dispersive component was expected, since the surfaces of the examined polymers are known as non-polar [40]. Even the presence of wood in PLA-W did not contribute to larger polar component of calculated SFE. Nevertheless, it needs to be taken into account that the calculated values of SFE are just a theoretical approximation on the basis of Owen–Wendt theory and empirical equations. Physical properties of surfaces of any kind of materials are dependent of many parameters, including CA measurement technique, substrates chemical properties, properties of tests liquids, surface morphology and zeta potential [41].

CAs of water droplets were somehow related to SFE. In general, the greater the polar part of SFE resulted in better wettability of surfaces with water. Here, all the surfaces turned out to be quite hydrophobic, with water CAs ranging from 72.8° (on PLA) to 125.4° (on sanded ABS) [41].

### 3.2. ATR-FTIR Spectra

The obtained ATR-FTIR spectra in the relevant wavenumber ranges and liquid coatings are shown in Figure 3. The results present a good basis for interpretation of chemical compatibility of the substrates and coatings.

The representative spectra of the solid substrates are shown in Figure 3a. In the spectrum of ABS, bands at 1602 cm^−1^, 1494 cm^−1^ and 1453 cm^−1^ are attributed to C–C stretching vibrations in the aromatic rings of the styrene part in ABS. Small peak at 966 cm^−1^ presents C-H bonds in trans-butadiene part of ABS, whereas peaks at 765 cm^−1^ and 700 cm^−1^ present CH_2_ group in vinyl-butadiene and styrene part of ABS [26,42,43]. Spectrum of PLA showed small peaks around 2754 cm^−1^ presenting C–H valence vibrations. Band at 1757 cm^−1^ presents C=O valence vibration, whereas characteristic peaks at 1333 cm^−1^, 1250 cm^−1^ and 1200 cm^−1^ are assigned to C–O–C vibrations [44]. In comparison to spectra of PLA, spectra of PLA-W showed the only difference in shoulder peak at around 1720 cm^−1^, which is assigned to C=O group present in wood flour [45,46,47].

ATR-FTIR was also effectively used to detect the chemical composition of a liquid coating formulation (Figure 3b). In C-WB the broad peak at around 3385 cm^−1^ is due to the stretching vibration of hydroxyl (–OH) and amino (–NH_2_) groups. The small peak at 2931 cm^−1^ is due to C–H stretching vibration. The single peak at 1731 cm^−1^ is the stretching vibration of C=O group peak, and finally, band at 1170 cm^−1^ indicates C–O stretching vibration [48,49]. C-SB, composed of alkyd resins and hydrocarbons as solvents, expressed the curve with characteristic peaks at 1770–1680 cm^−1^ presenting ester (C=O) groups, at 1360–1000 cm^−1^ presenting ester (C–O–C) groups and peak at 800–630 cm^−1^ presenting aromatic C–H and aromatic ring bending, respectively [50]. The spectra of C-ABS were shown to be quite different than spectra of solid ABS substrate. In comparison to spectra of ABS substrate, additional strong peaks at 1710 cm^−1^ (C=O stretching), 1360 cm^−1^ (O = H stretching), 1220 cm^−1^ (C–O stretching) appeared due to the presence of acetone in the sample [51].

### 3.3. Surface Color and Gloss

The mean values of color space parameters, determined on uncoated and coated non-sanded and sanded 3D-printed substrates, are listed in Table 1. It can be seen that after sanding (s) the substrates became lighter. By comparing the color of coated samples, the applied white C-WB gave the surfaces lighter (higher *L**) and slightly more blueish (higher *a**) appearance than white C-SB, regardless the type of 3D-printed substrate. These differences originate from different binders (acrylic or alkyd) and solvents (water or organic solvent) present in coating formulations. Sanding of the substrates prior to application of coatings also promoted lighter appearance (higher *L** values) of surfaces coated with C-SB and C-WB. Due to the original blue color of diluted ABS, the C-ABS obviously made the coated surfaces more blueish. However, the color of these samples was dependent on the color of the substrate, due to the partial transparency of the C-ABS.

The gloss of the surfaces is highly related to surface roughness (results presented in the next section). The gloss values detected at 60° angle of incidence are shown in Figure 4. After sanding (s), the surface gloss of 3D-printed substrates decreased (for about few G.U.), more significantly by 3D-printed PLA. Application of coatings increased the gloss of both non-sanded and sanded samples. In general, C-SB and C-ABS increased surface gloss more (both up to 85 G.U.) than C-WB (up to 36 G.U.). On the basis of these results, according to Zorll [52], the samples coated with C-SB and C-ABS can be denoted as gloss to high gloss, while the samples coated with C-WB can be denoted as mat gloss to medium gloss.

### 3.4. Surface Roughness

Results of mean surface roughness of uncoated and coated non-sanded and sanded (s) 3D-printed substrates, obtained on 2560 µm^2^ large area, are shown in Figure 5. The highest surface roughness was determined on non-sanded substrates, particularly PLA-W, followed by PLA and ABS. As reported by Yang and co-authors [53], the layer thickness and filling method have a significant effect on the surface roughness of the 3D-printed parts, while the printing speed has no effect on the surface roughness.

Surface roughness of 3D-printed substrates considerably decreased after sanding (even for more than 50% in case of PLA-W), and the differences in roughness between different types of substrate materials were reduced. Application of C-WB reduced surface roughness of non-sanded substrates (for up to 53%) but influenced surface roughness of sanded substrates less (reduced for up to 19%). On the contrary, however, the application of C-SB and especially the application of C-ABS reduced the surface roughness more obviously (for up to 78% in case on non-sanded PLA or for up to 79% in case of non-sanded ABS, respectively). The presence of acetone in coating made of diluted ABS probably at least partially also diluted the surface of coated polymer substrates. This is shown and discussed in the following sections. In general, the coatings with higher solid content are more effective at smoothing or planarizing the surfaces of 3D-printed products [54].

### 3.5. Microscopic Structure of Coated Polymers

The coatings film thicknesses were determined by observing the coated samples cross sections with the light microscope. Figure 6 shows two examples of the cross sections of 3D-printed materials with applied coatings. As shown in Figure 6a, the coating film thickness of either C-WB or C-SB can be relatively easily determined, also due to the high contrast in color between the coating film and the substrate (Table 2). However, this did not apply in case of the ABS coated with C-ABS, due to the undistinguishable transition between the substrate and coating (Figure 6b). Here, this brought a need to use other microscopic technique, namely SEM.

In general, C-SB formed films with greater thickness than C-WB, due to the higher solid content (70% compared to 45%). Thinner coating films were formed on sanded samples, which is related with lower surface roughness of sanded substrates in comparison to non-sanded ones.

SEM observations of ABS coated with C-ABS confirmed that the structure of the interface between the substrate and the coating is very homogenous (Figure 7a). For this reason, the interface with the transition between both materials cannot be found, and the coating film thickness and the coating penetration depth cannot be determined. Something similar was reported by Marciniak and co-authors [36] when observing ABS exposed to acetone vaporization with computed tomography, suggesting the in-depth dissolvent of ABS. The same authors report on impaired mechanical properties of the researched material.

The SEM micrograph of non-sanded PLA-W surface is shown in Figure 7b. The observations revealed the presence of wood dust particles embedded in the PLA polymer, which covers the surrounding surface area. The surface of PLA-W contains also many voids, formed during the printing process of samples. Such pores or voids often appear when 3D-priting with wood-plastic composites, due to the evaporation of water or other volatile compounds evaporating at extruding temperatures. Both, the presence of wood and the presence of micropores on the PLA-W surface most probably contributed to lower water CAs and enhanced coatings adhesion, as reported in the next section.

### 3.6. Coating Adhesion

Sanatgar and co-authors [55] report that the adhesion between different polymers can be explained by diffusion theory, where the chainlike molecules diffuse in the structure of counter polymer, which consequently leads to formation of a strong bond between adhesive and adherent. The authors also conclude that the polymer diffusion has a major effect on properties of the interface layer between the two polymers. Diffusion is, however, dependent on the temperature, composition, compatibility, molecular weight, orientation and molecular structure of polymers [56].

Measured coating films adhesion strengths differed among different coatings, as well as among different substrates (Figure 8). In general, the highest adhesion strengths were measured on samples coated with C-ABS and on PLA-W substrates. Better adhesion of C-ABS in comparison to other two types of coatings can be explained by findings from study of chemical composition (ATR-FTIR spectra) and microscopical study (SEM micrographs). In particular, the acetone presented in the coating formulation contributed to formation of a strong bond with the substrate polymers. Sanding of 3D-printed substrates seemed not to influence coating films adhesion. On both ABS substrates, the adhesion of C-ABS (slightly above 2.0 MPa) seemed to exceed the cohesive strength of the substrate. For this reason, the actual adhesion strength of coating cannot be defined. Coatings adhesion on PLA substrates showed similar values as on ABS substrates, with a difference that all the failures were of adhesion type. Here the highest adhesion was determined on samples coated with C-WB (2.2 MPa) and C-ABS (2.2 MPa), which was quite higher than by C-SB (1.5 MPa). The greater porosity of PLA-W surface was probably the main reason for the highest adhesion of all three types of coatings. The presence of voids, micro-pores and the wood particles presented on the surface of PLA-W seemed to enable more efficient penetration of the coatings in liquid state, better interlocking of coating and finally higher adhesion strengths of coating films. The highest adhesion was again determined on samples coated with C-ABS on both, non-sanded and sanded PLA-W substrates (2.9 MPa and 2.5 MPa). In both cases, a cohesive type of failure was determined, signalizing that the adhesion strength of the coating film was higher than cohesive strength of the substrate and therefore remains unknown.

It is worthwhile to mention that the mechanical properties of 3D-printed ABS and PLA are limited and, as determined with tensile and pull-off tests, can be enhanced with multi-material 3D-printing [57].

## 4. Conclusions

The samples produced with fused deposition modeling technique, using ABS, PLA and PLA-W filaments, expressed different wettability abilities. In general, the surface properties of 3D-printed polymeric substrates and their ability to interact with applied coatings are dependent on several aspects.

The calculated SFE of substrates have shown that all three types of the substrates have mostly non-polar character, regardless of whether they remain non-sanded or were sanded. The values of calculated total SFE (ranging from 29 mJ∙m^−2^ to 41 mJ∙m^−2^) and the CAs of water droplets (ranging from 72.8° to 125.4°) assigned the surfaces of all substrates as hydrophobic.

Recorded ATR-FTIR spectra showed that the substrates and coatings have very different chemical composition, which could alter their compatibility. The presence of wood in PLA-W only slightly differed from the spectra of PLA with additional peak at 1720 cm^−1^ showing the presence of C=O groups. This minor influence of wood in PLA on chemical composition of PLA-W polymer was probably also the reason for minimal differences in wettability properties between PLA and PLA-W substrates. Acquired ATR-FTIR spectra of coatings were mainly influenced on the type of solvent in formulation of particular coatings. The most presented –OH and –NH_2_ groups in C-WB make it more compatible with hydrophilic substrates, for instance with wood. C-SB composing of alkyd resins and organic solvents expressed the ATR-FTIR curve with esters and hydrocarbons. The spectra of C-ABS were mainly defined by the presence of acetone in coating formulation.

Color of coated 3D-printed substrates was dependent on mechanical preparation prior to surface treatment and the type of applied coating. Lighter hue of samples coated with C-WB than that of samples coated with C-SB originates in the coating formulations’ differences. Gloss of coated 3D-printed samples was increased after surface finishing with C-SB and C-ABS (both for about 80 G.U.) more than after surface treatment with C-WB (for 36 G.U.). Sanding decreased the surface roughness of 3D-printed substrates. Surface roughness of substrates was reduced after application of coatings, those in general confirming our hypothesis. Surface roughness was reduced more after surface treatment with C-SB and C-ABS than after surface treatment with C-WB.

Coating film thicknesses were found to be related to type of coating and substrate surface roughness. The sanding lead to formation of thinner films, while the films of C-SB were thicker due to the higher solid content. Microscopic study revealed that the C-ABS has the ability of diluting the substrate polymers due to the presence of acetone in formulation. Consequently, the interface between C-ABS and coated polymers had a homogenous structure where the transition between both materials cannot be found, and therefore, the coating film thickness could not be determined properly. However, this property positively influenced the adhesion strength of such surface systems. Another property contributing to better wettability and higher adhesion strengths of coatings was the presence of hydrophilic wood particles on the surfaces and higher porosity as was shown characteristically for PLA-W substrate.

Adhesion between substrate and coating depends on physical and chemical properties of both materials together, forming a surface system. The sanding process increased the smoothness of the surfaces, but not their porosity, which could contribute to better interlocking of the coating polymers. The presence of acetone in the coating formulation of C-ABS contributed to formation of a strong bond with the substrate polymers. The adhesion strength differed between the types of the substrates and the types of coatings, which confirms the second hypothesis posed at the beginning. The highest adhesion was on samples coated with C-ABS, followed by C-WB and the lowest by C-SB.

This study showed that the surface treatment of 3D-printed polymers is quite complex and depends on the properties of both, the substrate and coating. Certain visual and mechanical properties can be achieved with differently formulated coatings. However, the selection of the coating type depends on the required needs and desires of the potential user. Future research work on a similar topic could be focused on the other properties of different surface systems, including their monitoring during the exposure to periods of weathering or any other kind of influences.

## Figures and Tables

**Figure 1 polymers-12-02797-f001:**
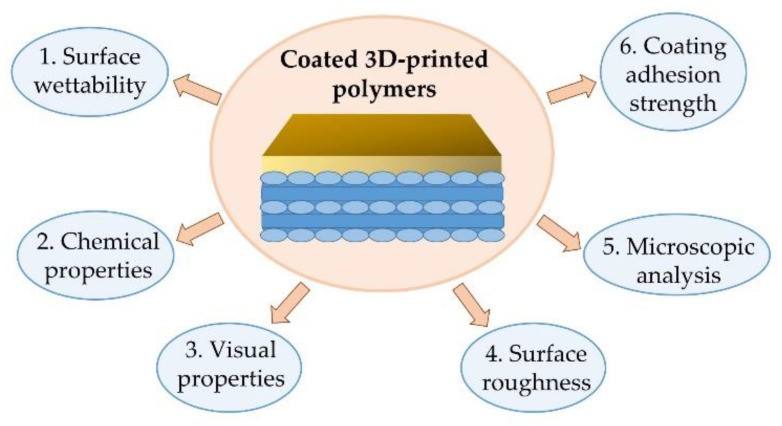
Sequential presentation of the study methodology.

**Figure 2 polymers-12-02797-f002:**
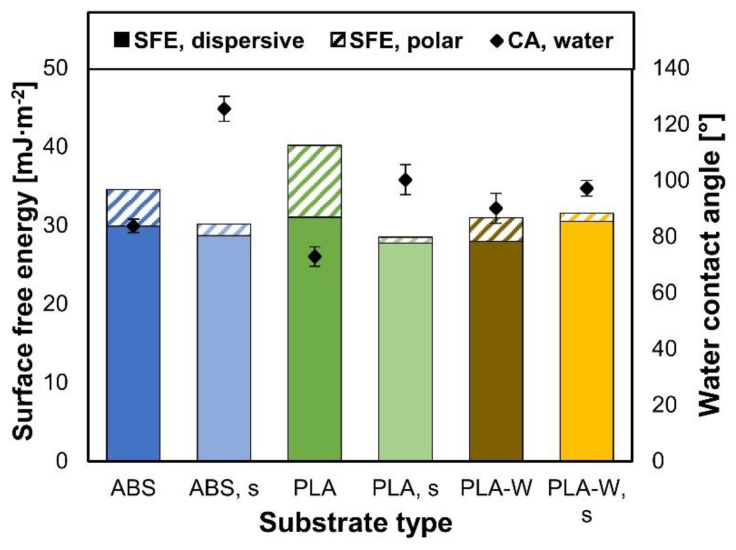
Calculated surface free energy of non-sanded and sanded (s) 3D-printed substrates, together with determined water CAs.

**Figure 3 polymers-12-02797-f003:**
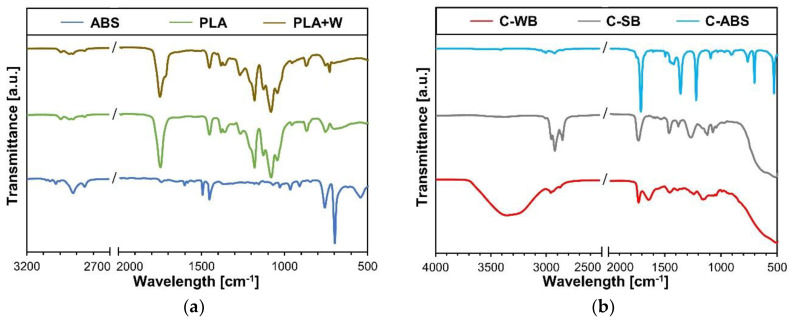
Acquired ATR-FTIR spectra: (**a**) Spectra of the 3D-printed substrates; (**b**) spectra of the coatings.

**Figure 4 polymers-12-02797-f004:**
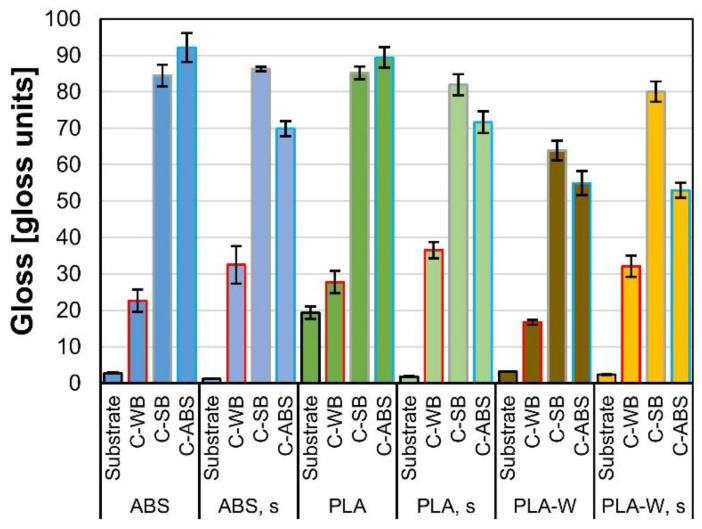
Gloss values of uncoated and coated non-sanded and sanded (s) 3D-printed substrates.

**Figure 5 polymers-12-02797-f005:**
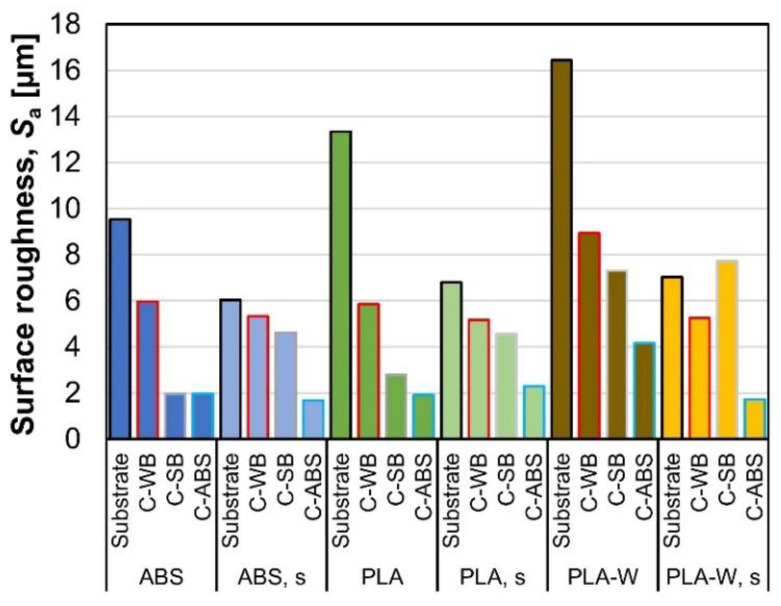
Mean surface roughness of uncoated and coated non-sanded and sanded (s) 3D-printed substrates.

**Figure 6 polymers-12-02797-f006:**
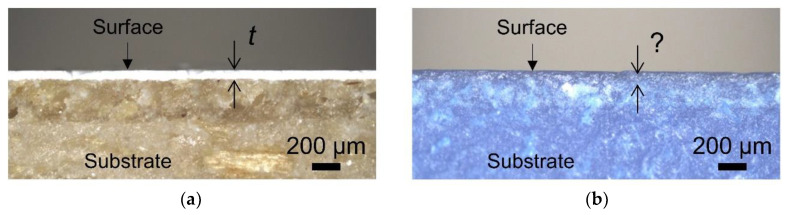
Cross sections of selected coated 3D-printed materials: (**a**) sanded PLA-W substrate coated with C-SB; (**b**) sanded ABS substrate coated with C-ABS.

**Figure 7 polymers-12-02797-f007:**
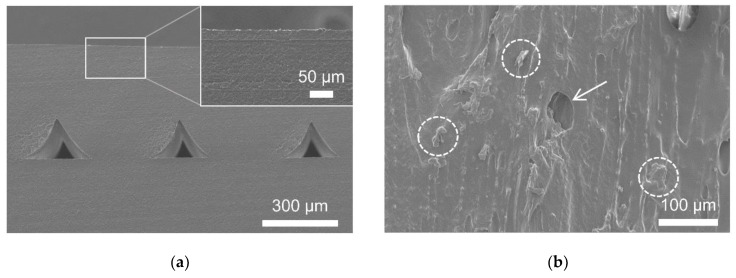
SEM micrographs showing the materials microscopical structure: (**a**) Cross-section of sanded ABS substrate coated with C-ABS. An inlay is showing a homogenous structure of the interface between the substrate and the coating, where no clear transition between those can be distinguished. (**b**) The microstructure of non-sanded PLA-W surface. The wood dust particles are encircled with a dashed line, whereas the arrow is pointing to the micropore presented on the substrate surface.

**Figure 8 polymers-12-02797-f008:**
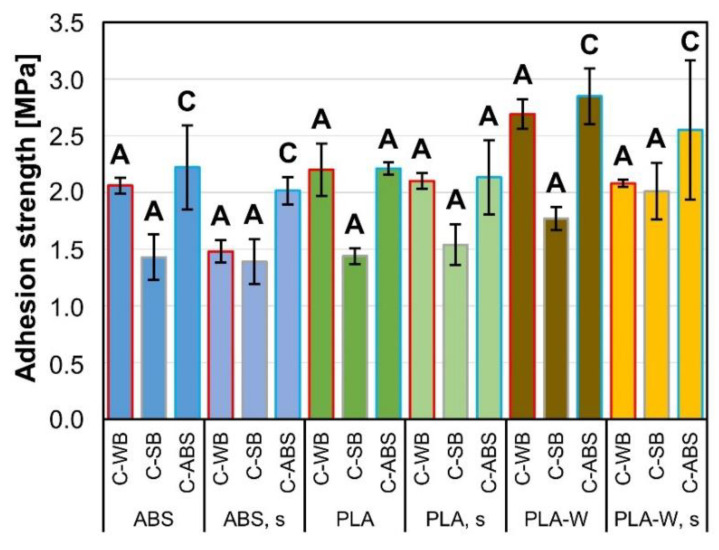
Adhesion strength of coatings on non-sanded and sanded (s) 3D-printed substrates, including the designation of the failure type: adhesion failure (A) of the coating film or cohesive failure (C) in a substrate.

**Table 1 polymers-12-02797-t001:** Color space parameters determined on uncoated and coated non-sanded and sanded (s) 3D-printed substrates.

Color Space Param.	Sample Type											
	ABS				PLA				PLA-W			
	Subst.	C-WB	C-SB	C-ABS	Subst.	C-WB	C-SB	C-ABS	Subst.	C-WB	C-SB	C-ABS
*L**	39.63	97.51	95.07	38.00	80.91	98.19	95.22	63.58	68.57	97.62	95.00	55.58
*a**	−7.51	−1.09	−1.27	−7.21	−1.12	−0.72	−1.12	−19.40	5.69	−0.92	−1.24	−15.00
*b**	−26.62	1.36	3.63	−26.30	1.90	2.28	3.72	−2.02	16.17	2.02	3.58	−7.87
	**ABS, s**				**PLA, s**				**PLA-W, s**			
	**Subst.**	**C-WB**	**C-SB**	**C-ABS**	**Subst.**	**C-WB**	**C-SB**	**C-ABS**	**Subst.**	**C-WB**	**C-SB**	**C-ABS**
*L**	43.84	96.97	94.84	40.43	81.43	98.1	95.28	67.04	70.24	97.09	94.68	60.71
*a**	−8.81	−1.27	−1.47	−8.09	−1.05	−0.76	−1.07	−17.50	5.11	−1.03	−1.29	−10.10
*b**	−24.72	0.83	3.15	−26.02	1.98	2.15	3.80	−17.10	14.88	1.56	2.71	−0.80

**Table 2 polymers-12-02797-t002:** Determined coating film thicknesses on non-sanded and sanded (s) substrates.

Film Thickness [µm]	Sample Type								
	ABS			PLA			PLA-W		
	C-WB	C-SB	C-ABS ^1^	C-WB	C-SB	C-ABS ^1^	C-WB	C-SB	C-ABS ^1^
Average	64.7	81.8	49.7	77.2	85.8	67.7	74.8	90.4	76
St. dev.	8	5.1	20.9	2.5	3.3	9.2	4.1	7.4	12.2
	**ABS, s**			**PLA, s**			**PLA-W, s**		
	**C-WB**	**C-SB**	**C-ABS ^1^**	**C-WB**	**C-SB**	**C-ABS ^1^**	**C-WB**	**C-SB**	**C-ABS ^1^**
Average	57.2	70.2	75.1	50.6	61.9	63.7	42.8	57.2	43.4
St. dev.	5.6	3.7	6.3	11.3	6.7	7.4	6.1	9.8	5.2

^1^ Harder distinguishable transition between the coating film and the substrate.

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
