# Peer review of "Surface Finishing of 3D-Printed Polymers with Selected Coatings"

_polymers, 2020, doi:10.3390/polym12122797_

Round 1

Reviewer 1 Report

GENERAL COMMENT

The aim of the present work was to assess the properties of three different substrates, 3D-printed of typical raw materials used for such purposes (ABS, PLA and PLA-W). The substrates were subsequently coated with three different coatings with the different type of solvent. The topic is in current interests for the readers.

The investigations would produce valuable outcomes, anyway the research presented the small issues:

Introduction

  • The state of the art correctly has been presented.

Materials and Methods

  • After what time, measurements of the contact angle were made?
  • Properties of the lacquer products (mainly commercial products) should be given.

Results

  • The investigation results were correctly and clearly presented.

Conclusions

  • Conclusions corresponds with the aim of the work. Further tests on the resistance of the coatings should be carried out.

References

  • The selection of the literature sources was correct.

Based on the above considerations my recommendation is: accept after minor additions.

Reviewer 2 Report

please see my comments in the attached file.
